# Malolactic Fermentation: New Approaches to Old Problems

**DOI:** 10.3390/microorganisms10122363

**Published:** 2022-11-29

**Authors:** Junwei Fu, Ling Wang, Jingxian Sun, Ning Ju, Gang Jin

**Affiliations:** 1School of Food and Wine, Ningxia University, Yinchuan 750021, China; 2School of Agriculture, Ningxia University, Yinchuan 750021, China; 3Engineering Research Center of Grape and Wine, Ministry of Education, Yinchuan 750021, China

**Keywords:** malolactic fermentation, lactic acid bacteria, global climate change, wine quality

## Abstract

Malolactic fermentation (MLF) is the decarboxylation of L-malic acid to L-lactic acid by lactic acid bacteria (LAB). For the majority of wine production, secondary fermentation is crucial. MLF significantly impacts the quality of most red and some white wine. The outcomes of the spontaneously initiated and finished MLF are frequently unpredictable and can even cause the wine to deteriorate. As a result, individuals typically favour inoculating superior starter cultures when performing MLF. The MLF method for wine has, however, faced new difficulties because of the altered wine fermentation substrate environment brought on by global climate change, the growing demands of winemakers for production efficiency, and the rising demand for high-quality wine. To serve as a reference for the study of wine production and MLF in the current situation, this review primarily updates and summarises the research findings on increasing the effectiveness and dependability of MLF in recent years.

## 1. Introduction

The complicated process of manufacturing wine involves two types of fermentation: yeast-led alcohol fermentation (AF), which primarily consists of converting sugar into ethanol and carbon dioxide. The second is malolactic fermentation (MLF), which is the process by which lactic acid bacteria (LAB) transform L-malic acid into L-lactic acid [1]. The majority of red wines and certain white wines are produced by using MLF, which is generally regarded as a secondary fermentation procedure. Along with deacidification, it can also alter the chemical and sensory characteristics of wine, increase biological stability, and decrease acidity [1]. One or more LAB species, primarily *Oenococcus*, *Lactobacillus*, *Pediococcus*, and *Leuconostoc*, are typically responsible for MLF induction. *Oenococcus oeni* is the most prevalent LAB species linked to MLF because it can adapt to the challenging wine conditions of high ethanol, low pH, high SO_2_ content, and low temperature [2]. However, since global climate change makes the alcohol getting higher and acid getting lower in wine (Figure 1), the habitat of bacteria is changing, it has recently been shown that *Lactobacillus* and *Pediococcus* can survive in different winemaking environments [3,4,5], and their metabolic processes play a more significant role in developing wine aroma and flavour [6,7]. But spontaneous MLF during winemaking is frequently unpredictable and challenging to watch over or manage, which could delay MLF and put the wine at risk. For instance, an unsteady bacterial habitat, an unpleasant odour, a lot of biogenic amines (BAs), and too much volatile acidity [8]. Wine microbiological stability and sensory quality depend on an effective and trustworthy MLF. Due to this, the primary goal of this paper is to look for a way to increase MLF effectiveness and wine quality under new conditions.

## 2. Effects of Different Inoculation Methods on MLF and Wine Quality

### 2.1. Co-Inoculation vs. Sequential Inoculation

Using commercial MLF starting cultures for fermentation has become the most popular way to make wine today. Determining the precise timing of inoculation is one of the elements to consider when utilising commercial starter cultures to induce MLF. These were co-inoculations (24 h after initiation of alcoholic fermentation) and sequential inoculations (after the end of the alcoholic fermentation) (Table 1). Contrary to sequential inoculations, potentially harmful interactions between yeast and bacteria during co-inoculation may contribute to fermentation failure. Additionally, the abundance of bacteria that can access sugar may lead to an increase in acetic acid [9,10]. To prevent harmful bacteria-yeast interactions and lower the danger of acetic acid generation due to low residual sugar content after AF, the majority of winemakers utilise sequential inoculation for MLF. In contrast, the co-inoculated starter completed MLF during AF, but wine fermentation with a sequential inoculation approach required 20 to 30 days to complete all fermentation tests [11]. This suggested that co-inoculation could shorten the overall fermentation period and effectively manage the MLF.

Different inoculation timings will also result in the release of various fragrance compounds in wine, modifying the qualitative and quantitative aspects of wine [23]. According to Izquierdo et al., co-inoculation increased the total acidity and lactic acid content of two different types of wine (Tempranillo and Merlot) more than sequential inoculation did. Additionally, it has more ethyl lactate and less tyramine. This study also showed that wines made via co-inoculation had reduced levels of several BAs, such as cadaverine and tyramine [17]. According to Abrahamse and Bartowsky, Shiraz wines fermented by co-inoculation and sequential inoculation have considerably different volatile component profiles. More fruity components were seen in wines under co-inoculation compared to sequential inoculation [24]. Another study offered evidence for the impact of inoculation time on the aroma and metabolic properties of Merlot wines produced in wineries. Co-inoculation, however, did not always promote the development of fruity scents, and it was shown that the strength of lactic aromas might either rise or fall in intensity [25]. Similar findings were made by Diez-Ozaeta et al., who found that co-inoculating yeast and bacteria could have a significant influence on wine’s volatile components and flavours. Their research revealed that, compared to sequential inoculations, the content of higher alcohols and acids in co-inoculated wines reduced dramatically [26]. According to Diez-Ozaeta et al., excessive amounts of acids and alcohol might disguise fruity and flowery fragrances and cause disagreeable odours [26]. As a result, the reduction of higher alcohol and acid content in co-inoculated wine would improve the quality. Additionally, ethyl lactate, isoamyl lactate, and diethyl succinate levels in all wines made with co-inoculated strains increased noticeably. The wines tasted good because these esters gave them fruity, creamy, and buttery qualities [27]. Apramita et al. also looked into how co-inoculation influenced the non-anthocyanin polyphenol profile of wine made from two distinct grape varieties. They found that the metabolic activities of LAB did not significantly impact phenolic compounds during co-inoculation [28]. On the other hand, Guzzon et al. suggested that co-inoculation would change the amount of tannins and anthocyanins in wine, reducing the astringency and improving the wine’s appearance [29,30]. Smit et al. concluded that the inoculation strategy also seems to affect the BAs content of the wines. The result of their study showed that less tyramine production was observed in co-inoculated Pinotage and Shiraz wines compared to wines inoculated with *Lentilactobacillus hilgardii* alone [31]. Together, these results show that co-inoculation significantly impacts wine quality and enhances fermentation efficiency.

### 2.2. Mixed Co-Inoculation

Non-*Saccharomyces* yeasts have recently been found to produce secondary metabolites during the MLF process which improved wine quality and increased the final flavour and texture of the wine. *Metschnikowia pulcherrima* and *Torulaspora delbrueckii*, two non-*Saccharomyces* yeast strains, were investigated by Balmaseda et al. for their impact on MLF in the production of red and white wines. Compared to wines infected with only *Saccharomyces cerevisiae*, the results demonstrated that wines inoculated with non-*Saccharomyces* yeast had higher anthocyanin concentrations and shorter MLF durations. Meanwhile, compared to *S. cerevisiae* alone, wine with inoculated *M. pulcherrima* and *T. delbrueckii* had decreased SO_2_ and medium-chain fatty acid contents. Additionally, it was found that *T. delbrueckii* metabolites appeared to contribute to the growth of *O. oeni* and enhance MLF function [32]. In addition, Zhao et al. inoculated *Saccharomyces cerevisiae*, *Pichia fermentans,* and *O. oeni* during the winemaking. It was found that *Pichia fermentans* contributed to the growth of *O. oeni* and MLF activity and also increased the fruit aroma of the wine [33]. However, most non-*Saccharomyces* yeast strains have a restricted capacity for fermentation, characterised by slow fermentation rates and poor SO_2_ and pH tolerance [19]. To improve the concentration of terpenoids, esters, higher alcohols, glycerol, acetaldehyde, acetic acid, succinic acid, and to reduce the yield of alcohol concentration, non-*Saccharomyces* yeast is typically blended with *S. cerevisiae* yeast [14]. Due to this, mixed inoculation tactics have recently attracted the attention of numerous researchers. To produce wines with the shortest MLF duration and improved sensory quality, co-inoculation of *S. cerevisiae* yeast, non-*Saccharomyces* yeast, and LAB was used during fermentation. According to Wang et al., whereas *O. oeni* strains reduced wine pyranoanthocyanin content during MLF, acetaldehyde generated by *Lactiplantibacillus plantarum* boosted pyranoanthocyanin accumulation in wines [34]. Devi et al. then looked at how wine’s anthocyanin concentration and colour were affected by combined inoculations of *L. plantarum* (Lp 1), *O. oeni* (Oo 1), and *S. cerevisiae* (AAV2). It was found that co-inoculation with the three strains could lessen the colour loss brought on by MLF [35]. Additionally, Tiziana et al. assessed the impact of several inoculation techniques and the inoculation of a non-*Saccharomyces* yeast, *T. delbrueckii*, in AF on the quality of Barbera wines. The outcomes demonstrated that the co-inoculation of *S. cerevisiae* yeast and non-*Saccharomyces* yeast did not affect the positive effects of LAB co-inoculation on the wine (such as fermentation time, chemicals, aroma, and sensory properties of wine), nor did it delay the AF and MLF processes. It also permitted the wines to acquire a more intense colour [36]. Although most research about mixed co-inoculations has been conducted on a laboratory scale, there are examples of industrial scale trials that showed the positive of mixed co-inoculation in achieving efficient winemaking. Maria et al. used *Candida zemplinina* (35NC1), *S. cerevisiae* (NP103), and *L. plantarum* (LP44) for industrial-scale winemaking and completed AF and MLF in seven days. Also, wines produced in this way contain higher concentrations of alcohols, esters, and terpenes, which add to the complexity of wine aroma [37].

### 2.3. Influence of Bacteria-Yeast Interactions on MLF

Wine flavour and quality were affected by the co-inoculation of *S. cerevisiae* yeast, non-*Saccharomyces* yeast, and LAB in a way that seemed connected to the strain and the inoculation time. According to Liu et al., LAB and yeast species can both have stimulating, inhibitory, or neutral effects on one another. These interactions mostly relate to how yeast can create metabolites that alter LAB metabolism from nitrogen compounds it consumes or releases [14]. The majority of studies have examined how yeast and LAB interact with each other. Tristezza et al. investigated how two yeast (CY1, CY2) and two *O oeni* (CL1, CL2) strains interacted with each other. It was determined that when the CY2 yeast strain and CL2 bacteria were injected simultaneously or sequentially, the levels of L-malic acid residues in wines were at their greatest. When they were co-inoculated, the CY1 yeast strain hampered the growth of the CL2 strain, whereas a delayed MLF happened when the bacteria were added after the AF. However, the presence of additional yeast and bacteria decreased the length of MLF and volatile acidity. As a result, the wine fermented in this way had an aroma of ripe fruit associated with esters, butter, and cream associated with diethyl succinate and ethyl lactate [38]. Russo et al. [39] found that when co-inoculated with LAB, yeast behaviour was unaffected during AF. In contrast, non-*Saccharomyces* yeast affects LAB differently depending on the species and strain, which impacts the length of the fermentation process and the generation of MLF metabolite components [40]. Fifty-two distinct yeast-bacteria combinations were studied by Nuria et al. for their impact on *O. oeni* and MLF. *T. delbrueckii*, *M. pulcherrima*, *Hanseniaspora uvarum*, *Hanseniaspora vineae*, and *Starmerella bacillaris* were five non-*Saccharomyces* yeasts. The results demonstrated that there were strain-specific differences in how non-*Saccharomyces* yeast affected *O. oeni* and MLF. Each strain of *H. uvarum* and *H. vineae* significantly raised the acetic acid concentration, which negatively affected the quality of the wine. The best circumstances for MLF were provided by specific *T. delbruecckii* and *M. pulcherrima* strains, which included little SO_2_ production, little consumption of L-malic acid, and enhanced mannoprotein concentration in the wine [22]. However, the results of this research cannot be used to explain the incomplete MLF in wines inoculated with *S. Bacillaris* 1109 and *O. oeni* PSU-1. It also demonstrates the intricacy of the parameters that affect the compatibility of strains and raises the possibility that other inhibitory substances have passive effects on yeast-bacterial interactions [22]. Du Plessis et al. investigated the interactions between *H. uvarum*, two species of *S. cerevisiae*, two LAB (*L. plantarum* and *O. oeni*), and two inoculation techniques in Shiraz wines. Only *H. uvarum* in wine was observed to inhibit MLF slightly after inoculation, which may be related to nutrient consumption or the production of poisonous compounds to LAB. But when *S. cerevisiae* and *H. uvarum* were combined, as opposed to *S. cerevisiae* inoculation alone, *H. uvarum* had a favourable impact on the growth of LAB [41]. This serves as further evidence of the multiple ways in which yeast-bacteria interactions affect the fermentation process. Furthermore, next generation sequencing (NGS) analysis was used for analysing the consist of microorganisms in Un-inoculated must, pied-de-cuve, *S. cerevisiae* alone, *S. cerevisiae,* and *T. delbrueckii* co-inoculation and sequential inoculation, *S. cerevisiae,* and *M. pulcherrima* co-inoculation and sequential inoculation. It’s interesting to note that different taxonomic compositions of the bacterial communities in the malolactic consortia, in terms of prokaryotic phyla and genera, result from each experimental scenario. They found that inoculation with different non-*Saccharomyces* (*M. pulcherrima*, *T. delbrueckii*) as well as the timing of inoculation affected the malolactic consortia and regulated MLF performance. For the first time, it was also demonstrated that inoculating the *M. pulcherrima* strain delayed MLF, whereas inoculating the *T. delbrueckii* yeast strain prevented MLF [42]. Many of the changes observed from co-inoculation are common to most of the yeasts and LAB. However, there are still certain variations that depend on the strain. An example is the amount of mannoprotein produced by the *Starm. bacillaris* [43]. Another example comes from Snyder et al. who found that certain strains of *Lachancea thermotolerans* with high lactic acid production inhibited the growth of *O. oeni* and the MLF process. The opposite result was obtained by using low lactic acid-producing strains [44]. Furthermore, some *S. cerevisiae* and non-*Saccharomyces cerevisiae* strains only strongly inhibited the growth and malolactic activity of the *O. oeni* strain [22], while having no effect on other strains. Therefore, choosing suitable yeast-bacterial strains is crucial to ensuring efficient and trustworthy MLF.

From the description above, it can be inferred that, when compared to sequential inoculation, simultaneous AF and MLF can result in faster and more consistent winemaking by carefully choosing compatible combinations of yeast-bacterial strains. Additionally, this method modifies the wine’s volatiles and sensory qualities, which impacts wine style. The interaction between yeast and LAB is still unknown, though. Future studies should uncover more information regarding the effects of yeast metabolism, including those of non-*Saccharomyces* yeast, on LAB and MLF performance.

## 3. New MLF Approaches to Handle Environmental Challenges in the Wine Industry

Since more than 30 years ago, winemakers have used *O. oeni* to induce MLF while avoiding specific *Pediococcus* and *Lactobacillus* strains. *Lactobacillus* strains have been linked to spoilage during slow AF and several wine deterioration symptoms, including a musty aftertaste and excessive acetic acid. As some *Pediococcus* strains produce exopolysaccharides that increase wine viscosity, they are typically regarded as spoilage bacteria in wine. Recent research has revealed that some *Pediococcus* and *Lactobacillus* do not taint wine and may even enhance its quality. As a result, using these strains to induce MLF increased the variety of MLF starter cultures and led to the production of more distinctive wines (Table 2). Therefore, the use of these strains to induce MLF not only expanded the biodiversity of MLF starter cultures, but also produces more characteristic wines.

### 3.1. Selection of MLF New Strains

#### 3.1.1. The Performance of *Lactobacillus* spp. in MLF

##### *Lactiplantibacillus* *plantarum*

*L. plantarum* is frequently present in wine and contributes to spontaneous MLF. The wine industry has received much attention lately as a beginning culture that can initiate MLF in place of *O. oeni* [58,59]. Due to the impact of climate change in recent years, most wines now have increased pH and ethanol levels, wines with high ethanol result in slow or unsuccessful MLF, while high pH levels are more likely to develop spoilage microbes, which can lower their sensory quality and cause the formation of poisonous substances such as EC and BAs [60]. Since *L. plantarum* is a homo-fermentative bacteria, it only creates lactic acid while metabolising glucose and not acetic acid, reducing any potential risk from an excess of volatile acid [52]. *L. plantarum* has therefore been suggested for the biological acidification of wines [47,61]. Urbina and Lucio et al. co-inoculated *L. plantarum*, *O oeni*, and yeast strains, respectively, to control wine acidity. They concluded that wines MLF with *L. plantarum* fermented more quickly and contained more lactic acid than wines inoculated with *O. oeni* [62,63]. *L. plantarum* was further demonstrated by Jiang et al. to enhance wine aroma while acidifying wine. By employing low-acidity Cabernet sauvignon for pilot-scale winemaking, they assessed the acidification and fermentation capabilities of *L. plantarum* in two inoculation procedures. According to the findings, both *L. plantarum* inoculation techniques generated a significant amount of lactic acid, which raised the titratable acidity of Cabernet Sauvignon wine. Additionally, there is a large increase in the concentration of lactate-related esters (ethyl lactate, ethyl butanoate, ethyl hexanoate, etc.), giving the wines a “fruit” and “butter” scent [64]. These investigations show that *L. plantarum* can be used to cause the biological acidification of wine. The growing demand of current consumers for better, healthier foods has generated new interest in the study of BAs. In recent years, approaches to reduce BAs in wine have been focused on the use of degradable BAs microorganisms. Among the LAB associated with wine, *L. plantarum* showed higher BAs degradation activity compared to other strains. Two *L. plantarum* strains (NDT09 and NDT16) from Italian wins were examined by Capozzi et al. They demonstrated that they did not form BAs and could simultaneously break down putrescine and tyramine [65]. Recently, an *L. plantarum* strain with BAs degradation and wine environment stress resistance was screened. It was inoculated and carried out MLF in Carmenere wine. The result should that the content of histamine, tyramine, and cadaverine in wine was decreased by more than 57% after MLF. Additionally, the glyceraldehyde-3-phosphate dehydrogenase (GAPDH) that could break BAs down in this strain was isolated, and it was found that its activity remained stable at high pH (5.5–8.5) or high temperatures (30 °C–50 °C) [66], reflecting an excellent anti-stress quality of GAPDH. Moreover, Jiang et al. successfully expressed GAPDH in Escherichia coli for the first time and purified it. It was also found that recombinant GAPDH was highly resistant under both sulphate and ethanol stress, and the histamine in wine was broken down by 83.1–85.5% [67].

*L. plantarum* is crucial for inducing MLF and enhancing the sensory qualities of wine, in addition to biological acidification and the breakdown of BAs. According to reports, both *L. plantarum* and *O. oeni* can resist the harsh environment of wine [48,52,68]. For instance, Bravo et al. used the inoculation of particular strains to assess the quantities of malic acid ingested in a wine-like media. Eight *L. plantarum* isolates were found to be completely able to take malic acid after only four days of inoculation [69]. Brizuela et al. also found that some *L. plantarum* strains identified from Patagonian wines were capable of MLF at lower inoculum levels than *O. oeni* strains and did not require previous domestication treatments [70]. *L. plantarum* also has a larger variety of enzymes that can continue to function under winemaking conditions [13]. Mtshali et al. looked into whether *L. plantarum*, which was isolated from South African wines, contained any genes encoding for enzymes with oenological significance. Unexpectedly, they found several genes that code for enzymes, including β-glucosidase, protease, esterase, citrate lyase, and phenolic acid decarboxylase. According to these results, *L. plantarum* may not only be able to trigger MLF but also serve as a possible source of enzymes that enhance wine scent. But further research is required on how these genes are expressed when the wine is present [71]. A study by Takase et al. showed that *L. plantarum* is capable of biotransforming S-3-(hexan-1-ol)-L-cysteine (3SH-S-cys) and S-3-(hexan-1-ol)-L-cy-steinylglycine (3SH-S-cysgly) to produce 3SH [72]. It contributes significantly to the aromatic complexity of the wine. However, neither the conversion-causing enzymes nor putative regulatory mechanisms have been identified. Therefore, in addition to focusing on their ability to consume malic acid, it is important to understand how the genes encoding these aroma-related enzymes are controlled during winemaking circumstances when employing *L. plantarum* strains to induce MLF, to assess their degrees of expression and enzymatic activity under various winemaking circumstances to enhance the sensory qualities and quality of the wine.

As mentioned above, *L. plantarum* is not only capable of surviving in challenging wine environments, but also doesn’t create acetic acid when metabolizing glucose. It has various enzymes that can create more sophisticated aromatic molecules. On the other hand, some strains can prevent bacterial spoilage and the breakdown of BAs. These qualities give *L. plantarum* the potential to become the MLF starter culture in the future.

##### *Lentilactobacillus* *hilgardii*

Previously, it was believed that *L. hilgardii*, primarily isolated from wine, is the bacteria that causes wine deterioration. According to reports, *L. hilgardii* produces harmful metabolites such as BAs and EC. Recent research has shown that, like other LAB species, its capacity to make BAs and EC precursors cannot be generalised but is very strain-dependent [73]. some strains create fewer or none, others produce more toxic metabolites [31,74]. When metabolising malic acid under controlled circumstances, the superior LAB that was intentionally chosen can produce just a small amount of hazardous chemicals. The LAB will metabolise additional components of wine, such as amino acids and tartaric acid, if they are not managed and eliminated promptly after MLF. This will result in hazardous compounds (BAs, EC, etc.) or microbe diseases. In addition, the amount of BAs produced by *L. hilgardii,* as suggested, seems to be related to LAB interactions. when *L. hilgardii* and *O. oeni* were combined, compared to *L. hilgardii* inoculation alone, *L. hilgardii* increased the histamine content in the wine-make medium by 34% [75]. According to Darya et al., *L. hilgardii* differs from other bacteria in the genus *Lactobacillus* in that its genome has undergone remarkable reorganisations, and it has a wider variety of adaptation skills [50]. According to Gustaw et al., *L. hilgardii* is resistant to sulphates, acidic conditions, and relatively high ethanol concentrations [51,76]. It is also a possible source of new esterase because it contains genes for various metabolic processes, and some enzymes are active in winemaking settings [52]. From the wine-making medium, Sumby et al. successfully isolated the esterase EstC34 of *L. hilgardii* CSCC-5489. Under the circumstances of wine, this esterase remained stable and functional. Additionally, it can decrease short-chain ethyl esters like ethyl acetate and is crucial for wine fragrance [77]. It has been found that most of the aromatic substances in wine (esters, monoterpenes, C13-noisoprene, benzene derivatives, etc.) exist in the form of glycosides binding state, which is not easy to be perceived. Therefore, in order to enhance and improve the aroma of wine, it is necessary to convert the binding aroma precursor substances into free. β-glucosidase is the key enzyme in the release of the binding aroma [33,78]. Seven *L. hilgardi* strains obtained from Albarino wine were investigated by Lopez et al. for their *hdc*, *odc*, *tdc*, and *pad* genes. They found that some strains could not synthesise BAs and had not yet begun to create volatile phenols, which could harm the scent. Additionally, every *L. hilgardii* strain demonstrated has the capacity to generate β-glucosidase at various doses [74]. More research is necessary to assess their appropriateness to start MLF and their behaviour in pilot-scale and commercial-scale fermentations. The level of alcohol in wine has been rising over the past few years due to concerns about global warming. As a result, the high ethanol percentage may cause the MLF process to be slow or possibly stop. As a result, it is crucial for the wine industry to choose starter cultures for MLF that are highly ethanol tolerant. Three *L. hilgardii* strains were identified from Australian Grenache wines by Jin et al. They found that all three cultures could thrive in MRS-AJ medium with 19% (*v*/*v*) ethanol [51]. These investigations point to the potential of this species as a starter culture for the next generation of malolactic bacteria.

#### 3.1.2. *Pediococcus* spp. Has MLF Potential Strains

A group of bacteria with high nutrition requirements is called *Pediococcus*. Some strains have been isolated from wines worldwide, including *P. damnosus*, *P. inopinatus*, *P. parvulus*, and *P. pentosaceus*. However, they have long been regarded as wine-spoiling organisms [56]. For instance, Landate et al. demonstrated that the *Pediococcus* isolate could produce a significant amount of BAs, which would cause wine degradation [79]. Some strains can produce exopolysaccharides (EPS), which are β-glucans made of D-glucose, according to Llaubères et al. These polymers give the wine a “sticky” flavour and boost its viscosity [80,81]. Additionally, *Pediococcus* can convert glycerol into acrolein, which reacts with the phenolic compounds in wine to give it a severe “bitterness” [82]. However, some research indicates that the presence of *Pediococcus* doesn’t always bring on wine degeneration. According to Edwards and Peterson et al., some *P. parvulus* strains can produce major alcohol and may change the aroma of wines [83]. Enzymatic activities of 85 LAB strains, which will influence the degradation of histamine, putrescine, and tyramine, were compared by Garcia et al. The result showed that nine of the strains were the most capable of degrading BAs and they belonged to the genera *Lactobacillus* and *Pediococcus* [84]. Callejon et al. found that *Pediococcus acidilactici* CECT5930 could break down histamine, tyramine, and putrescine in a synthetic medium and wine. A multiplex oxidase was isolated and purified from the strain to serve as the amine degradation enzyme [85]. Additionally, Juega et al. noted that *P. damnosus*-induced MLF in high-pH Albarino and Caino wines. In addition to not producing BAs or EPS, it gave the wines flowery and fruity smells [86]. Additionally, applying *Pediococcus* in high-pH wines decreased the addition of SO_2_ and gave wines with less effective SO_2_ added some microbiological stability. According to this, they may be used as malolactic starting cultures in high-pH wines. Additionally, a study indicated that Pinot Noir wines infused with *P. parvulus*, *P. inopinatus*, or *P. damnosus* had higher concentrations of flowery and red fruit scents. It also prevents *Brettanomyces bruxellensis* from producing 4-ethyl phenol [57]. Although *Pediococcus* strains have been demonstrated in several recent studies to have favourable effects on wine, isolates and species can have different sensory effects. This shows that the species and strain present directly impact the variety of sensory responses generated by *Pediococcus* in wine and the formation of BAs and EPS. Therefore, future research should focus on comprehending the variations between strains and the interactions between the genus and other bacteria. Perhaps its position in winemaking could be revisited, and it might be considered the next generation of malolactic starting culture.

### 3.2. Research on the Adaptation of Wine Strains to Environmental Variations

Numerous techniques enhance LAB strains to produce more trustworthy and effective MLF, most notably genetic and non-genetic recombination. Recombination methods often focus on inserting or deleting particular genes with high accuracy. Kaur et al. investigated the variation of phenolics in wine that finish MLF by using recombinant *Pediococcus acidilactici* BD16 (*fcs*^+^/*ech*^+^) for MLF. It was found that the concentration of vanillin and numerous other cinnamic acid derivatives increased significantly, enhancing the aroma, color, and flavor of the wine [87]. This strategy is typically more constrained because customers do not generally support using genetically modified organisms in the food sector. Utilizing mutagens is another way to enhance genetic traits. Various chemical or physical factors, such as UV light, were used to be mutagens. Li et al. successfully screened *O. oeni* C10-1 using UV mutagenesis to improve the quality of MLF. It was found that the malolactic conversion rate of *O. oeni* C10-1 was 38.81% higher than the original strain. The fermented wine had the highest color, which alleviated the decolouration of the wine after MLF [88]. However, mutagenesis introduces random changes to the DNA that could result in the loss of advantageous traits. The requirement to thoroughly screen the populace following the application of mutagens has also hindered it. Directed Evolution (DE), the spontaneous adaptability to high-stress conditions, is an example of non-genetic recombination. Some organisms could pick up advantageous mutations that allow them to flourish, multiply, and finally take control under particular circumstances [89]. To create *O. oeni* A90 with greater ethanol tolerance, Betteridge et al. first employed the DE method, and the parent strain *O. oeni* SB3 was continually grown in an MRS-AJ medium with rising ethanol content (5–15%). It was shown that in a medium with high ethanol concentration, an isolate from this strain could complete MLF earlier than its parent *O. oeni* SB3 after about 350 generations [90]. Next, to generate *O. oeni* strains capable of more efficient MLF. Jiang et al. inoculated *O. oeni* A90 in a red wine-like environment with increasing levels of stressors for about 350 consecutive generations. Three strains were selected from the DE process based on their ability to fermentate in the Red Fermented Chemically Defined Grape Juice Medium (RFCDGJM)/wine which was blended with 15.1% (*v*/*v*) ethanol, 26 mgL^−1^ SO_2_ at pH 3.35. All three strains consumed more L-malic acid than the parent strain, which was stuck. Further studies evaluated the fermentation performance of the three superior strains screened in four red wines (pH 3.37–3.55; ethanol 13.9–16.7% (*v*/*v*)). It was found that they induced faster MLF rates in the wines and that the optimized strains produced reduced or equal amounts of acetic acid compared to the commercial strains in Mouvedre and Merlot wines [91]. These findings prove that using DE under winemaking conditions can improve *O. oeni* performance and MLF.

### 3.3. Other Strategies

An essential part of an organism’s antioxidant system is the tripeptide antioxidant glutathione (GSH). It keeps proteins functioning normally and supports LAB’s ability to withstand various conditions, including oxidative stress, osmotic stress, acid stress, and cold stress [92]. Numerous studies have connected *O. oeni*’s increased GSH reductase to its ethanol adaption [93]. Even though *O. oeni* cannot synthesize GSH, some researchers claim that it contains genes capable of metabolizing GSH [92,94]. Seven genes were identified as involved in the GSH redox system in *O. oeni* PSU-1, including the peroxidase (*gpo*), three glutaredoxin-like genes, a reductase (*gshR*), and two subunits for the hypothetical GSH transporter, *cydC* and *cydD*. Margalef et al. measured the relative expression of these seven genes. The results revealed that just a few of these genes responded to elevated GSH, including a potential glutamatergic oxydoxin and the putative glutathione transporter component *cydC*. After pre-adaption in a wine-like medium, evaluation of the *O. oeni*’s malolactic performance revealed that the addition of GSH during pre-adaption growth protected against cells exposed to low pH and ethanol, leading to a faster MLF [95]. Su et al. looked into the impact of *O. oeni* SD-2a which was pre-adapted in a GSH-supplemented medium on the quality of cabernet franc wine. Her research showed that the presence of GSH enhanced *O. oeni* SD-2a growth under acidic (pH 3.2) and alcoholic (12% (*v*/*v*)) stress conditions. After the completion of MLF, the total phenol, flavonol, flavanol, proanthocyanidin, syringic acid, and trans-resveratrol concentrations in wine were increased, improving the antioxidant potential of the wine. Additionally, it enhances the redness and colour saturation of wine [96]. L-proline supplementation during the growth of *O. oeni* increased its ethanol tolerance and MLF performance, according to Catello et al. [97]. Although these modifications boosted *O. oeni*’s tolerance to ethanol and acid stress, they also improved MLF, which improved wine quality. However, the mechanisms at play are unclear.

Pulsed electric field (PEF) technology has currently been demonstrated as a key technique to enhance malolactic fermentation performance by successfully inactivating several common microorganisms in wines [98,99]. To lessen the pressure that *O. oeni* faces from the competition with other microbes, Lucia et al. use PEF after AF to minimise the wine’s microbial population [100]. This technique shortened the duration of MLF. A different study by Lucia et al. found that PEF technology improved the sensory quality of the wine in addition to causing a modest reduction in MLF time. The wine treated with it has less volatile acidity and a more intense colour [101]. The use of PEF techniques throughout the winemaking process has not yet been studied, and most PEF experiments have been done in static settings. Utilizing immobilised cells that may be recycled for numerous cycles is another method for enhancing MLF performance. Encapsulation of bacterial cells and adsorption to the carrier have been the main fixing techniques employed [102,103]. Simo et al. have studied the effects of encapsulating *O oeni* in SiO_2_-alginate hydrogel (Si-ALG) biocapsules on LAB stability and cell contraction during winemaking [104].

In addition, it’s critical to keep an eye on the MLF procedure as wine is being made to enhance wine quality. Recent devices for measuring MLF include wireless sensor bungs, ultrasonic sensors, and others [105,106]. A modern electrochemical bienzymatic biosensor with great sensitivity and low detection limits was created by Gimenez et al. To measure L-malic acid; the sensor relies on the co-interception of MDH and DP enzymes with redox mediators in an electro synthetic PPy membrane. The sensor preserved more than 90% of its sensitivity after 37 days when it was used to measure MLF in wine [107]. This is solid proof that winemakers may employ biosensors to keep an eye on MLF fermentation and maintain the quality of their product.

## 4. Conclusions

People now have a better understanding of how MLF impacts wine quality because of ongoing studies. To better regulate MLF, various solutions have also been created in response to the increasing issues of new challenges. For instance, the timing of LAB strain inoculation, mixed strain fermentation, PEF technology, and immobilised cells can improve wine aroma without releasing too much acetic acid and reducing the duration of MLF. Although little is known about the interaction between yeast and bacteria, we still don’t understand why the distribution of aromatic chemicals is altered during the fermentation of mixed strains. Future research in genomes and metabolomics may be utilised to explain the many metabolites produced throughout the MLF process and how they relate to yeast, laying the groundwork for improving comprehension and management of MLF.

Additionally, consumers do not now permit or approve of the use of genetically engineered organisms. Due to this, more outstanding research on the LAB biodiversity in wine will be helpful for either developing strains that are excellent at regulating wine taste and aroma compounds or for using directed evolution to develop novel strains that are more suited for growing in wine-growing environments. Glutathione and L-proline can also be added to the medium to increase the tolerance to the wine environment of the strains, enhancing MLF and wine quality. At the same time, it is also important to monitor MLF in real-time through technologies such as biosensors. This will not only allow winemakers to manage the process more efficiently, but also reduce the incidence of delays in wine production.

## Figures and Tables

**Figure 1 microorganisms-10-02363-f001:**
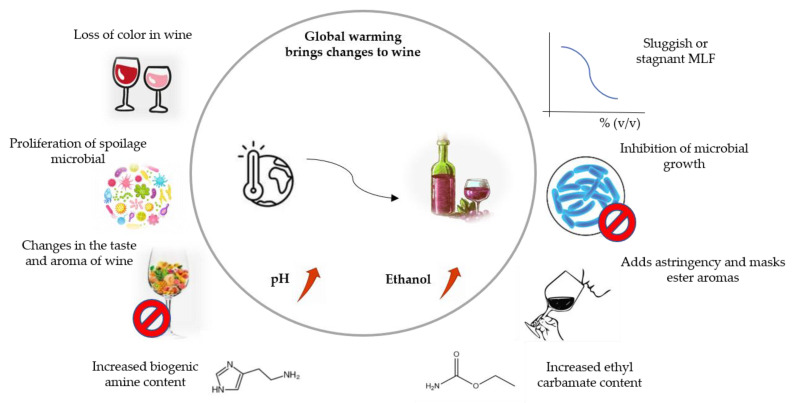
Effects of climate change on wine quality.

**Table 1 microorganisms-10-02363-t001:** Comparison of the strategies of sequential inoculation and Co-inoculation to improve for MLF.

Method	Sequential Inoculation	Co-Inoculation
Positive effects	Yeast produces nutrients available for LAB [12] Avoid the production of acetic acid and D-lactic acid [13] Prevents adverse interactions between LAB and yeast [14]	Improve the efficiency of MLF [15] Enhance wine aroma complexity [16] Reduce the concentration of BAs in wine [17] Reduces color loss as well as bitterness and astringency perception in wines [12] Prevents the growth of *Brettanomyces* and produces phenolic off-odors [18] Metabolites produced by yeast strains promote LAB growth and improve MLF performance (Strain dependence) [19]
Negative effects	Prolong or delay of MLF [20] Increase the risk of microbial spoilage [7] Produces undesirable compounds such as BAs and Ethyl carbamate (EC) [7]	Inhibition of MLF [21] Sluggish or stagnant AF [14] Increased acetic acid content, which adversely affects the quality of the wine [22]

**Table 2 microorganisms-10-02363-t002:** New malolactic bacteria and their role in MLF.

Strain	Past	Present
*L. plantarum*	Causes wine spoilage, including mousy off-flavours and excess acetic acid [2]	BAs degradation [45] Involved in improving wine color [34] With the most diverse range of plantaricins [46] Induced biological acidification of wine [47] More complex enzyme profile (Glycosidase/Protease/ Esterase/Lipase/Citrate lyase) [48]
*L. hilgardii*	Can produce undesirable me-tabolites such as BAs and EC (strain dependence) [49]	Have a wider range of environmental adapt-ability [50] MLF starter cultures with high tolerance to ethanol [51] Can be used as a source of new esterases to improve wine aroma (strain dependence) [52]
*Pediococcus* spp.	Causes wine spoilage, including ropiness and high level of diacetyl [53,54] Produce BAs and EC (strain dependence) [55]	Provide enzymes that degrade BAs [6] Induction of MLF in high pH red and white wines [56] May offer the opportunity to develop novel wine styles [57]

## Data Availability

Not applicable.

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
