# Peer review of "Malolactic Fermentation: New Approaches to Old Problems"

_microorganisms, 2022, doi:10.3390/microorganisms10122363_

Round 1

Reviewer 1 Report

1. Current section 2 is large consisted with long paragraphs. I suggest split section 2 into subsections, with carefully chosen subsection titles to organize the content. 

2. It seems Table 1 does not include all the pros and cons of sequential inoculation versus co-inoculation mentioned in the text. 

3. While NGS was mentioned at the end of page 4, there was no description of any of the findings. I suggest the authors expand the findings from NGS.

4. I am not convinced how climate change would affect the wine production. Isn't wine fermentation usually implemented under well controlled bioreactors?

Author Response

请参阅附件

Reviewer 2 Report

The paper “Malolactic fermentation: New approaches to old problems” summarized recent studies on increasing the effectiveness and dependability of wine MLF. The manuscript is novel and well organized. Co-inoculum (S. cerevisiae/LAB and S. cerevisiae/non-Saccharomyces/LAB), new MLF starters, and the strategies to improve LAB are highlighted. The information and perspectives in this paper are quite helpful to ensure effective and reliable wine MLF.

Minor comments and suggestions:

Keywords

“new approaches” is too general, please change.

Introduction

Page 1 “Oenococcus oeni is the most prevalent LAB species… the challenging wine conditions” Please specify which conditions.

Section 2

Page 2 “Contrary to sequential inoculations, …yeast and bacteria may contribute to fermentation failure.” Dose this occur in co-inoculations? Please specify.

Page 3 “Different inoculation times will also result in the release of…”. Using “inoculation timings” may be proper.

Other non-Saccharomyces yeasts like Lachancea thermotolerans, Pichia fermentans and Candida zemplinina are also important for the management of MLF. Please summarize their roles in affecting LAB growth, MLF activity and flavor outcomes. (For example, doi: 10.20870/oeno-one.2021.55.2.4657, doi: 10.1016/j.foodres.2022.111604, doi: 10.3390/fermentation3040064)

Section 3

Names of LAB in the titles of section 3.1.1 (3.1.1.1 and 3.1.1.2), and section 3.1.2 should be given in italic font, including “Lactobacillus”, “Lactiplantibacillus plantarum”, “Lentilactobacillus hilgardii” and “Pediococcus”.

3.1.1.1 Please specify which lactate-related esters in “Additionally, there is a large increase in the concentration of lactate-related esters”. Please revise the statement since L. plantarum can also produce acetic acid: “L. plantarum is not only capable of surviving in challenging wine environments, but also doesn't create acetic acid.”

3.1.1.1 or 3.1.1.2 Given that β-glucosidase is important flavor enzyme produced by non-Oenococcus LAB, please introduce the role of this enzyme in the aroma formations of wines.

Tables

Table 1 Replace “positive effects” with “Positive effects”.

Table 2 Replace “Pediococcus spp” with “Pediococcus spp.”.

References list

Some article title should be corrected according to their pdf versions. For example, reference 11, 60 and 76.

Besides, too many “-” occurred in the title, for example, “Lactobacillus plantar-um” (8), “Wine m-icrobi-ome” (14), “O-ccurrence” (52). Please check throughout the list.

Round 2

Reviewer 1 Report

All my concerns have been addressed.